# Assessment of the Effectiveness of a Computerised Decision-Support Tool for Health Professionals for the Prevention and Treatment of Childhood Obesity. Results from a Randomised Controlled Trial

**DOI:** 10.3390/nu11030706

**Published:** 2019-03-26

**Authors:** George Moschonis, Maria Michalopoulou, Konstantina Tsoutsoulopoulou, Elpis Vlachopapadopoulou, Stefanos Michalacos, Evangelia Charmandari, George P. Chrousos, Yannis Manios

**Affiliations:** 1Department of Dietetics, Nutrition and Sport, School of Allied Health, Human Services and Sport, La Trobe University, Melbourne, VIC 3086, Australia; 2Department of Nutrition and Dietetics, Harokopio University of Athens, 70 El Venizelou Avenue, Kallithea, 17671 Athens, Greece; mariamichal95@gmail.com (M.M.); kontsou@gmail.com (K.T.); manios@hua.gr (Y.M.); 3Department of Endocrinology-Growth and Development, Children’s Hospital P. A. Kyriakou, 11527 Athens, Greece; elpis.vl@gmail.com (E.V.); stmichalakos@gmail.com (S.M.); 4Division of Endocrinology, Metabolism, and Diabetes, First Department of Pediatrics, University of Athens Medical School, Aghia Sophia Children’s Hospital, 11527 Athens, Greece; echarmand@med.uoa.gr (E.C.); chrousge@med.uoa.gr (G.P.C.); 5Division of Endocrinology and Metabolism, Center of Clinical, Experimental Surgery and Translational Research, Biomedical Research Foundation of the Academy of Athens, 11527 Athens, Greece

**Keywords:** personalised, nutrition, intervention, children, obesity, healthcare professionals

## Abstract

We examined the effectiveness of a computerised decision-support tool (DST), designed for paediatric healthcare professionals, as a means to tackle childhood obesity. A randomised controlled trial was conducted with 65 families of 6–12-year old overweight or obese children. Paediatricians, paediatric endocrinologists and a dietitian in two children’s hospitals implemented the intervention. The intervention group (IG) received personalised meal plans and lifestyle optimisation recommendations via the DST, while families in the control group (CG) received general recommendations. After three months of intervention, the IG had a significant change in dietary fibre and sucrose intake by 4.1 and −4.6 g/day, respectively. In addition, the IG significantly reduced consumption of sweets (i.e., chocolates and cakes) and salty snacks (i.e., potato chips) by −0.1 and −0.3 portions/day, respectively. Furthermore, the CG had a significant increase of body weight and waist circumference by 1.4 kg and 2.1 cm, respectively, while Body Mass Index (BMI) decreased only in the IG by −0.4 kg/m^2^. However, the aforementioned findings did not differ significantly between study groups. In conclusion, these findings indicate the dynamics of the DST in supporting paediatric healthcare professionals to improve the effectiveness of care in modifying obesity-related behaviours. Further research is needed to confirm these findings.

## 1. Introduction

A plethora of epidemiological data reports the high prevalence of obesity, an “epidemic” that represents a huge public health burden for many countries. Besides the increased risk for chronic diseases, obesity is also related to nutrient insufficiencies, a paradox that has been characterised as the “double burden of malnutrition” [1]. This “double burden” paradox can be interpreted by the existence of a chronic, low-grade inflammation state that is produced and sustained in obese children [2], leading to low blood concentration of essential micronutrients, such as iron [3] and vitamin D [4]. Considering the important roles of these micronutrients in several cellular, metabolic and physiological processes, their long-term insufficiency in obese individuals may become detrimental for children’s optimal growth and development. 

Due to the huge dimensions and detrimental effects of obesity and related complications, these conditions have been the major focus of public health research over the past decade. However, existing tools, programmes and strategies to counteract the “obesity epidemic” have only experienced limited success [5]. This is mainly due to the inadequate understanding of the complex mosaic of mechanistic pathways leading to obesity. In this regard, excess body weight is not only the product of a positive energy balance, but also an interaction of a plethora of other etiological factors, such as environmental ones that exert their effects even from very early life stages, such as the prenatal period and the first 5 years of life. By acting “in utero” (e.g., maternal obesity, smoking during pregnancy, etc.) or during infancy (infant formula feeding, growth velocity, etc.), perinatal factors can cause permanent endocrine adaptations, usually expressed as increased hunger, adipogenesis and consequently obesity at later life stages [6,7]. Another important reason for the limited or only short-term effectiveness of weight management programs is usually their delayed implementation in already obese children or in adulthood, when the energy-balance-related behaviours (EBRBs) and consequently the obesity phenotype are already established [8]. As such, the implementation of intervention initiatives as early in life as possible, when EBRBs and their determinants are still flexible, is promising for the prevention of obesity and related cardiometabolic complications [9]. 

Health professionals (i.e., general practitioners, family doctors, paediatricians, dietitians, nutritionists) have a key role among health experts, in prospectively and frequently monitoring children [10,11]. Furthermore, this key role places them into a central position, with regards to childhood obesity prevention and treatment, since they are also the ones guiding parents in providing the appropriate healthcare to their children. However, these professionals on many occasions they require additional and appropriate support to conduct a thorough assessment and provide tailor-made diet and lifestyle optimisation advice to families with children in need of weight management [12,13]. 

As such, the objective of this study was to examine the effectiveness of a computerised decision-support tool (DST), developed to assist paediatricians and paediatric endocrinologists in delivering personalised nutrition and lifestyle optimisation advice to children and their families, as a means of childhood obesity management. 

## 2. Materials and Methods

### 2.1. Development of the Decision Support Tool

The development of the computerised DST is based on decision-tree algorithms (Appendix A provides an example of these algorithms), which include five different levels, namely the “assessment of children’s current weight status” (level 1), the “assessment of the likelihood for the future manifestation of obesity in normal-weight children” (level 2), the “evaluation of the most appropriate body weight management goal” (level 3), the “estimation of children’s dietary energy and macronutrients intake needs” (level 4) and the delivery of “personalised diet and lifestyle optimisation advice” (level 5). 

The first level of the decision tree algorithms (“assessment of children’s current weight status”) is based on the measurement of body weight in all age groups from infancy to adolescence and of the recumbent length in infants and children until the age of 2 years or standing height in all children and adolescents after the age of 2 years. The international Body Mass Index (BMI)-for-age growth curves and the relevant reference values proposed by the WHO are further used to finalise the assessment of children’s weight status [14] and categorise them into “underweight” (BMI-for-age < 5th percentile), “normal weight” (5th percentile ≤ BMI-for-age < 85th percentile), “overweight” (85th percentile ≤ BMI-for-age < 95th percentile) and “obese” (BMI-for-age ≥ 95th percentile). 

The second level of the decision tree algorithms (“assessment of the likelihood for the future manifestation of obesity in normal-weight children”) is important because even if a child’s current body weight is normal, this does not exclude the likelihood of the future manifestation of obesity, especially in children that are subjected to the combined effect of obesity risk factors. In an attempt to examine the likelihood for the future occurrence of obesity in normal weight children, due to the combined effect of individual obesity risk factors, including socio-demographic and perinatal ones, the CORΕ (Childhood Obesity Risk Evaluation) index [15], was included as another component of the DST. More specifically, the CORE index represents a simple, easy-to-use and valid score [16], which provides an estimation of the future likelihood of obesity manifestation as early as the age of 6 months. This estimation achieved through the combined use and scoring of easily collected data on specific perinatal risk factors, such as maternal pre-pregnancy weight status, maternal smoking during pregnancy, infant’s weight gain during the first 6 months of life, as well as simple socio-demographic indices, namely the child’s gender and mother’s educational level. 

In the third level (“evaluation of the most appropriate body weight management goal”) the decision tree algorithms use the recommendations of the American Pediatric Association as a basis for the prevention and treatment of child and adolescent overweight and obesity [17]. More specifically, data collected on children’s age and current weight status, as well as on the presence of obesity-related comorbidities (i.e., hyperglycaemia, insulin resistance, dyslipidaemia, hypertension) in children and of obesity in one or both parents, are combined to inform each one of the following weight management pathways: (i) body weight maintenance, which aims to the progressive reduction of BMI due to the increase in height stemming from children’s growth, or (ii) body weight loss, whenever this is deemed appropriate, such as in cases where comorbidities and/or parental obesity co-exist with childhood obesity.

Following the evaluation of the most appropriate weight management goal, the fourth level of the decision tree algorithms (“estimation of children’s dietary energy and macronutrients intake needs”) is necessary to facilitate weight maintenance or weight loss as well as children’s growth. The mathematical formulas provided by the Institute of Medicine (IOM) for infants, children and adolescents [18] were used to assess estimated energy requirements (EER). After the estimation of dietary energy intake requirements, the DST calculates the percent distribution of energy into macronutrients, within the Acceptable Macronutrient Distribution Ranges (AMDRs) proposed by the IOM for carbohydrates, fat and protein for infants, children and adolescents [18].

In the fifth level (“personalised diet and lifestyle optimisation advice”), the decision tree algorithms analyse all aforementioned data and deliver a report providing the assessment of the examined child, as well as body weight, diet and lifestyle recommendations that will support the decision of health professionals. The report includes (a) the assessment of children’s current weight status and the need for body weight maintenance or loss, (b) the assessment of the likelihood for the future manifestation of obesity in normal-weight children, (c) children’s total dietary energy requirements based on the anticipated body weight management (i.e., weight maintenance or loss) target, (d) children’s dietary needs in carbohydrates, total fat and protein, (e) personalised meal plans, as well as (f) diet and lifestyle optimisation recommendations, tailored to the specific needs and weight management goals set for each child. The recommendations include practical advice to the family on how (i) to achieve an energy and nutrients’ balanced diet, via an increase in the consumption of foods that are rich sources of dietary fibre and complex carbohydrates and a reduction in the consumption of foods that have a high content of simple sugars, total and saturated dietary fat, cholesterol and sodium, (ii) to become more physically active, (iii) to reduce sedentary activities and (iv) to improve children’s sleep patterns [19].

### 2.2. Operational Components of the DST

The DST comprises of two operational components, namely the data entry and the data processing component. Regarding data entry, paediatric healthcare professionals collect information on the child’s gender and birth date and conduct anthropometric measurements of body weight, recumbent length or standing height (depending on the child’s age). Healthcare professionals also collect perinatal, socio-demographic and parental data, as well as some additional information on characteristics related to the child. In terms of perinatal factors, data is collected on maternal pre-pregnancy body weight (in kg), maternal smoking habits during pregnancy, while the child’s health record is used to copy information with regards to the child’s weight (in kg) at birth and at six months of age. Regarding socio-demographic and parental data, information is collected on self-reported mother’s educational level (in years of education), and on measured mother’s and father’s body weight (in kg) and height (in cm). Furthermore, healthcare professionals use a set of validated questions [20] to collect appropriate data that will allow them to categorise the child’s physical activity level, into light (<4 METs), moderate (4–7 METs) or vigorous (>7 METs). Lastly, information on the presence of obesity-related comorbidity indices, such as insulin resistance, dyslipidaemias and hypertension is also collected, either based on the child’s physical examination or based on biochemical or clinical indices from the child’s medical record that is available to the paediatric healthcare professionals. 

As far as data processing is concerned, all data are uploaded to the DST, which processes them and extracts a report with the child’s assessment and the personalised diet and lifestyle optimisation recommendations. More specifically, the DST uses the birth and examination dates to calculate child’s age (in months and years), it then calculates child’s BMI (in kg/m^2^) and consequently estimates the child’s weight status, through its categorisation into underweight, normal-weight, overweight or obese. In normal-weight children, the DST also calculates the CORE index score, based on which children with a higher (i.e., CORE index score ≥ 4) likelihood for obesity manifestation in childhood or adolescence are identified [16]. In addition, the DST calculates the estimated dietary energy requirement (in kcals per day) for the child, so as to achieve the desired body weight management (i.e., weight maintenance or loss) goal, while relevant calculations are also made with regards to dietary protein, carbohydrates and fat needs (in grams per day). Furthermore, the DST processes the data uploaded for parents, thus calculating parental BMI (in kg/m^2^) and categorising parents as non-obese or obese (i.e., BMI > 30 kg/m^2^). Finally, the DST proposes diet and lifestyle optimisation advice recommendations for the child and/or the entire family (Appendix A provides examples of the recommendations), as well as personalised weekly meal plans adjusted to the estimated energy requirements calculated for each child (Appendix A provides examples of the meal plans).

### 2.3. Personalised Lifestyle Optimisations Recommendations and Weekly Meal Plans

The DST follows five steps dictated by the decision tree algorithms (Appendix A provides the relevant steps) to propose personalised lifestyle optimisation recommendations and weekly meal plans. 

In step 1, children are categorised based on their BMI into normal-weight, overweight or obese, while in step 2 the CORE index score is calculated for normal-weight children. In normal-weight children with a lower likelihood for the future manifestation of obesity, the DST proposes diet and physical activity recommendations, which support the maintenance of normal body weight and growth (recommendation 1). 

In step 3, the DST focuses on normal-weight children with a higher likelihood for the future obesity manifestation and evaluates the co-existence of clinical disorders (i.e., hyperglycaemia, insulin resistance, dyslipidaemia and/or hypertension). In normal-weight children with no clinical disorders and with non-obese parents, the DST advises health professionals to provide recommendation 1 (i.e., similar to step 2 above). In normal-weight children with no clinical disorders but with at least one obese parent, the DST advises health professionals to provide specialised recommendations, aiming to improve diet and physical activity habits for the entire family (recommendation 2). In normal-weight children with at least one clinical disorder but with non-obese parents, the DST provides recommendations, aiming at maintaining the child’s normal body weight, but also delivering practical advice that supports the consumption of foods rich in dietary fibre and complex carbohydrates, but simultaneously the reduction in the consumption of foods high in simple sugars, total and saturated fat, dietary cholesterol and sodium (recommendation 3). Finally, in normal-weight children with at least one clinical disorder and with at least one obese parent, the DST provides recommendations targeting the entire family and aiming to improve physical activity and dietary habits for all family members (recommendation 4). The DST also proposes a periodic re-evaluation every 6 months for high-risk normal-weight children with at least one clinical disorder and/or at least one obese parent and every 12 months for children with no clinical disorders and/or non-obese parents. 

In step 4 the DST focuses on overweight children. In overweight children with no clinical disorders and with non-obese parents, the DST advises health professionals to provide recommendation 1, but also an isocaloric weekly meal plan, aiming to maintain the child’s body weight (meal plan 1) and consequently to progressively decrease its BMI (as the child grows and height increases), ideally below the 85th percentile. In overweight children with no clinical disorders and at least one obese parent, the DST provides recommendation 2, that targets the entire family, as well as the isocaloric meal plan 1, which aims for the maintenance of the child’s body weight. In overweight children with at least one clinical disorder and with non-obese parents, the DST proposes recommendation 3, as well as an isocaloric meal plan (meal plan 2), aiming for the maintenance of the child’s body weight via the consumption of foods rich in dietary fibre and complex carbohydrates, but also with a lower content of simple sugars, total and saturated fat, dietary cholesterol and sodium, compared to meal plan 1. Finally, in overweight children with at least one clinical disorder and with at least one obese parent, the DST advises health professionals to provide recommendation 4 to the entire family, as well as meal plan 2. The DST also suggests a periodic re-evaluation every 3 months for overweight children with at least one clinical disorder and/or at least one obese parent and every 6 months for children with no clinical disorders and/or non-obese parents. If the re-evaluation shows no reduction of BMI below the 85th percentile, the DST follows the same process described under Step 4. If the re-evaluation shows a reduction of BMI below the 85th percentile, the DST follows the process described under Step 3.

In step 5, the DST focuses on obese children. In the case of 2–5-year-old obese children, the DST follows exactly the same approach dictated by Step 4 for overweight children. The main differentiation occurs in 6–15-year-old obese children to whom mild weight loss is also prescribed. In this regard, in 6–15-year-old obese children with no clinical disorders and at least one obese parent, the DST targets the family and proposes recommendation 2 and a hypocaloric meal plan (meal plan 3). In 6–15-year-old obese children with at least one clinical disorder and non-obese parents, the DST proposes recommendation 3, as well as a hypocaloric meal plan (meal plan 4), via the consumption of foods rich in dietary fibre and complex carbohydrates, but also the decrease in the consumption of foods rich in simple sugars, total and saturated fat, dietary cholesterol and sodium. Finally, in 6–15-year-old obese children with at least one clinical disorder and with at least one obese parent, the DST targets the family and proposes recommendation 4 and a hypocaloric meal plan 4. The DST also proposes a periodic re-evaluation every 3 months for obese children with at least one clinical disorder and/or at least one obese parent and every 6 months for children with no clinical disorders and/or non-obese parents. If the re-evaluation shows no reduction of BMI below the 95th percentile, the DST follows the same approach described under Step 4 or Step 5, depending the child’s age (i.e., 2–5 or 6–15 years old). If the re-evaluation shows a reduction of BMI below the 95th percentile, but BMI remains higher than the 85th percentile, the DST follows the pathway dictated by Step 4. If the re-evaluation shows a reduction of BMI below the 85th percentile, the DST proposes the process described under Step 3. 

Table 1 summarises the target population and the behavioural change goals and lifestyle optimisation advice provided by each level of recommendations through the DST.

### 2.4. Randomised Controlled Trial to Assess the Effectiveness of the Computerised DST

The effectiveness of the DST was assessed through a pilot randomised controlled intervention trial (RCT). The RCT was initiated on May 2018 and was conducted in the Endocrinology Department of the “P. and A. Kyriakou” Children’s Hospital and in the Division of Endocrinology, Metabolism, and Diabetes of the “Aghia Sophia” Children’s Hospital in Athens, Greece. Before the study initiation, a statistical power calculation indicated that a total sample size of 64 children (50% females) would be adequate to observe a mean BMI difference of 1.5 kg/m^2^ between the two study groups (statistical power of 80% and level of statistical significance at 5%). Taking into account an attrition rate of 20%, a screening conducted in the premises of the aforementioned settings managed to recruit a total sample of 80 children, who were identified as eligible to be included in the RCT. The main eligibility criteria for inclusion in the RCT were children aged 6–12 years old, as well as overweight or obese status (i.e., BMI-for-age ≥ 85th percentile). Signed informed consent forms were obtained from all parents of eligible children, before their participation to the study. The study was conducted in accordance with the rules of the Declaration of Helsinki of 1975, revised in 2013 and the protocol was approved by the Bioethics Committee of Harokopio University, Athens (approval no.: 61/30-3-2018). Finally, the RCT was registered to clinicaltrials.gov (NCT03819673). 

### 2.5. Study Groups

The 80 overweight or obese children that were eligible to participate in the RCT, were randomly and equally allocated to two study groups. Those children that were randomly allocated to the intervention group (IG), were examined by paediatricians (i.e., general paediatricians and paediatric endocrinologists) and a dietitian, who were all trained in the use of the DST. A manual of operation with detailed instructions on the use of the DST was prepared and distributed to medical practitioners prior to the commencement of the study. The dietitian also assisted the paediatricians to assess children’s weight status, to set appropriate weight management goals and to provide personalised meal plans and/or recommendations to children and their families. In contrast, those families whose children were randomly allocated to the control group (CG), were provided with general recommendations of diet and physical activity and follow-up appointments were made for weight checks. The effectiveness of the intervention was evaluated through the collection of data at baseline and at a follow-up examination after 3 months. 

### 2.6. Data Collection: Parental Socio-Demographic and Anthropometric Characteristics

Data on specific socio-demographic characteristics were collected from parents (most preferably from the mother) during the scheduled face-to-face interviews. All interviews were conducted by the paediatricians or the dietitian with the use of a standardized questionnaire. The socio-demographic data collected by parents included father’s and mother’s age, educational level (years of education) and occupation. In addition, parents also reported or had their body weight and height measured, from which BMI was calculated and used to categorise each parent based on their weight status.

### 2.7. Dietary Intake

Dietary intake data were obtained by the dietitian with the use of a 24-h recall of one typical day in terms of children’s dietary intake and with a short food frequency questionnaire (FFQ), via interviews conducted with parents of children younger than 10 years of age or directly with children older than 10 years old. 

According to the data recorded from the 24h-recall, all study participants were asked to describe the type and amount of foods and beverages consumed, during the previous day, provided that it was a typical day according to the participant’s perception. To improve the accuracy of food description, standard household measures (cups, tablespoons, etc.) and food models were used to define amounts. At the end of each interview, the dietitian reviewed the collected data with the respondent in order to clarify entries, servings and possible forgotten foods. Dietary intake data were analysed using the Nutritionist V diet analysis software (version 2.1, 1999, First Databank, San Bruno, CA, USA), which was modified to include traditional Greek dishes and recipes [18]. Furthermore, the database was updated with nutritional information of processed foods provided by independent research institutes, food companies and fast-food chains. 

In addition, a short semi-quantitative valid FFQ [21] was used to collect data on children’s dietary intake of foods representing all main food groups (i.e., fruits, vegetables, grains, dairy and protein foods). The FFQ included questions that evaluate the consumption frequency of foods during the previous 3 months in frequencies ranging from less than 1 portion/month to more than 4 portions per day.

### 2.8. Perinatal Data

Regarding perinatal data, mothers were asked to recall information on their pre-pregnancy body weight and smoking practices during pregnancy. Additionally, mothers were asked to report their child’s body weight and recumbent length at birth and 6 months of age, as this was recorded at their child’s health record.

### 2.9. Physical Activity Levels

Organised and leisure time physical activities were assessed using a standardized questionnaire, that was also used and validated in the multicentre Feel4Diabetes study that was conducted in six European countries, including Greece [20]. Respondents reported the type, time (in minutes) and frequency (in times per week) spent by children on organised and/or leisure time physical activities.

### 2.10. Anthropometric Data

Body weight was measured to the nearest 0.1 kg using a digital weight scale (Seca Alpha, Model 770, Hamburg, Germany). Subjects were weighed without shoes in minimal clothing. Height was measured to the nearest 0.1 cm using a commercial stadiometer with subjects not wearing shoes, their shoulders in a relaxed position, their arms hanging freely and their head aligned according to the Frankfort plane. Weight and height were converted to BMI using Quetelet’s equation (weight (kg)/height^2^ (m^2^)), while the international BMI-for-age growth curves and the relevant reference values proposed by the WHO [14] were issued to calculate BMI z-score. Waist circumference (WC) was also measured to the nearest 0.1 cm with the use of a non-elastic tape and with the child standing, at the end of a gentle expiration. The measuring tape was placed around the trunk, at the level of the umbilicus, midway between the lower rib margin and the iliac crest.

### 2.11. Statistical Analysis

Normality of the distribution of continuous variables was analysed using the Kolmogorov-Smirnov test. Normally distributed continuous variables were expressed as Mean values (+/−Standard Error of the Mean: SEM) and categorical variables were reported as frequencies (%). Associations between continuous and categorical variables were examined using Student’s *t*-test for normally distributed variables or the non-parametric Mann-Whitney test for skewed variables even though logarithmic transformations were made. The associations between categorical variables were assessed using the chi square (χ^2^) test. Repeated-measures ANOVA was used to evaluate the significance of the differences among study groups at baseline and at the 3-month follow-up (treatment effect), the significance of the change from baseline to follow-up observed within each group (time effect) and the treatment × time interaction effect. The between-group factor was the study groups (i.e., IG compared to CG) and the within-group factor was the time point of measurement. Adjustments were also made for potential possible confounding factors. All reported *p*-values were based on two-sided tests. The level of statistical significance in all analyses was set at *p* < 0.05. The SPSS vs. 24.0 (SPSS Inc., Chicago, IL, USA) software was used for all statistical analyses.

## 3. Results

From the initial total sample of 80 children randomly allocated to the two study groups, 15 children (5 from the IG and 10 from the CG) could not be re-examined at follow-up. Figure 1 provides the flow diagram of the study according to the CONSORT guidelines.

The attrition resulted in a total sample of 65 children (35 in the IG and 30 in the CG) with full data at baseline and follow-up. The descriptive characteristics of these children and their parents at baseline are summarised as mean (+/−SEM) or as percentages in Table 2**.** Regarding demographic indices, the mean age of children participating in the study was 9.7 (0.2) years, while the mean age of fathers and mothers was 46.1 (0.3) and 41.2 (0.3) years, respectively. Furthermore, 24.6% of mothers had <9 years of education, which is the compulsory education level in Greece, while 42.6% had a higher education of >12 years. Regarding behavioural indices, the mean dietary energy intake recorded for children was 1535.6 (81.3) kcal per day with the percentage of energy coming in a descending order from carbohydrates (47.4%), fat (35.4%) and protein (18.5%), while the mean daily time spent by children on physical activity was 21.6 (2.3) min. As far as perinatal indices were concerned, the mean birth weight and recumbent length of children was 3.2 (0.1) kg and 50.7 (0.4) cm, respectively, while mean maternal pre-pregnancy BMI was 24.9 (0.4) kg/m^2^, with 15.5% of mothers being obese before conception. Regarding anthropometric indices, children’s mean body weight, height, BMI and WC was 51.9 (1.9) kg, 142.4 (1.4) cm, 25.1 (0.5) kg/m^2^ and 79.9 (1.5) cm, respectively, with 60.7% of children being obese. In addition, the mean BMI of fathers was 28.6 (0.4) kg/m^2^, with 27.6% of them being obese, while the mean BMI of mothers was 27.3 (0.4) kg/m^2^, with 31.6% of them being obese. Regarding differences between study groups, the mean BMI of mothers of children in the CG was higher than that of mothers of children in the IG (28.9 (1.2) vs. 26.0 (0.8) kg/m^2^; *p* = 0.045). No other statistically significant differences were observed between study groups.

The mean (SEM) values at baseline and follow-up examination, as well as the mean (95% CI) changes from baseline to follow-up, for both study groups with regards to children’s dietary intake of energy, macro- and micro-nutrients are presented in Table 3. Regarding dietary energy intake, no significant differences were observed between groups regarding the changes from baseline to follow-up, despite the decrease observed in the IG and the increase in the CG. As far as macronutrient intake was concerned, the increase observed in the IG for dietary fibre intake (4.1, 95% CI: 1.4 to 6.8) was higher than the non-significant change recorded in the CG (*p* = 0.005). In addition, sucrose intake decreased significantly only in the IG (−4.6, 95% CI: −8.8 to −0.3), although no significant differences were observed between study groups. Regarding micronutrient intake, significant increases were observed in the IG for iron (2.6, 95% CI: 0.2 to 5.0), zinc (1.7, 95% CI: 0.1 to 3.3) and magnesium intake (36.6, 95% CI: 9.4 to 63.8). In the case of magnesium, the significant increase observed in the IG was also higher than the change observed in the CG (*p* = 0.011). Lastly, a significant decrease was observed for vitamin C intake in the CG (−28.4, 95% CI: −53.6 to −3.1), although no group difference was found with regards to the changes from baseline to follow-up. No other significant changes within groups or differences between study groups were observed in the dietary intake of the rest of macro- and micro-nutrients, despite the fact that some of the changes were more favourable in the IG than the CG (e.g., for calcium, potassium, sodium, vitamin A and vitamin D).

Table 4 depicts the changes in the consumption of specific food items and the relevant differences between the two study groups. More specifically, children in the IG had a higher mean consumption of cereals at follow-up than children in the CG (0.78 (0.11) vs. 0.43 (0.12), *p* = 0.041). In addition, the consumption of yogurt decreased significantly only in the CG (−0.23, 95% CI: −0.42 to −0.50), while the consumption of chocolates (−0.32, 95% CI: −0.52 to −0.11), cakes (−0.13, 95% CI: −0.23 to −0.02) and chips (−0.08, 95% CI: −0.13 to −0.03) decreased significantly only in the IG. The changes observed for the consumption of yogurt (*p* = 0.005) and chocolates (*p* = 0.025) were significantly different between the two study groups.

The changes from baseline to follow-up, as well as the differences between study groups with regards to anthropometric indices are presented in Table 5. Body weight and WC increased significantly only in the CG by 1.4 kg (95% CI 0.3 to 2.6) and 2.1 cm (95% CI 0.7 to 3.5), respectively, height increased significantly in both study groups by 2.0 cm (95% CI 1.5 to 2.5) in the IG and by 1.6 cm (95% CI 1.0 to 2.1) in the CG, while BMI and BMI z-score decreased significantly only in the IG by 0.4 kg (95% CI −0.9 to −0.1) and 0.2 standard deviations (−0.3 to 0.05). Nevertheless, these changes were not found to differentiate significantly between the two study groups.

## 4. Discussion

The current randomised controlled trial showed that a computerised DST designed to assist paediatric healthcare professionals in providing personalised nutrition and lifestyle optimisation recommendations to overweight or obese children and their parents, can result in favourable changes to certain dietary intake and anthropometric indices in the children that received the intervention. The findings of this study support the growing, although still limited, body of evidence regarding the effectiveness of computerised or eHealth DSTs used in primary care settings for improving clinicians’ performance on childhood obesity management outcomes [22,23].

Health professionals have the potential to influence large numbers of patients. Up to date there has been little evidence on how clinical practice can be enhanced in order to assist children (and their parents) in achieving appropriate to their weight status and sustainable weight management. The role of new technology, through the development of appropriate computerised or e-Health tools, seems to be the way forward. Although there are currently several computerised or e-Health tools designed to promote personalised advice on weight management in children, the vast majority of those do not involve health professionals in the implementation process [24]. Even in the case of e-Health tools that are targeting health professionals, in most of the occasions their usability has been described as difficult [22,24]. As such, in the HopSCOTCH Shared-Care Obesity Trial in Australia, the general practitioners (GPs) that used the relevant e-Health tool to deliver the personalised intervention to children and their parents, characterised implementation as challenging and usability of the tool as poor, mainly due to technical reasons, such as out-dated hardware, software installation difficulties and poor internet connections [22].

Despite the scarcity of tools supporting paediatric healthcare professionals on children’s weight management, Taveras et al. [23,25] developed a computerised tool very similar to the DST developed in the current study. The effectiveness of this tool was examined in the “Study of Technology to Accelerate Research” (STAR), which was a three-arm, cluster-randomised controlled trial that was implemented in 14 paediatric offices in Massachusetts and on 800, 6 to 12-year-old, obese children [25]. After 12 months of intervention, the STAR trial reported a lower increase in BMI in children randomised in the study group that received the personalised advice via the use of the DST by paediatric healthcare professionals compared to the control group that received the usual care offered in the participating paediatric offices (mean adjusted BMI change difference: −0.51 kg/m^2^; 95% CI −0.91 to −0.11) [23]. The aforementioned results of the STAR study agree with the findings of our study, which -although they included a smaller sample size of 65 children and had a shorter duration of 3 months- reported a mean adjusted BMI change difference of −0.6 kg/m^2^ in the IG, compared to the CG. Similarly to the STAR trial, the effect of the intervention implemented in the current study on BMI also exceeded the mean adjusted change difference observed in other primary-care intervention trials, such as the “Live, Eat and Play” (LEAP) study (mean adjusted BMI change difference: −0.20 after 9 months) [26], the LEAP-2 study (mean adjusted BMI change difference: −0.11 after 12 months) [27] and the “Shared-Care Obesity Trial in Children” (HopSCOTCH) study (mean adjusted BMI change difference: (−0.10 after 12 months) [28]. In addition to BMI, the significant increase in waist circumference observed only in the CG is another indication of the effectiveness of the current RCT in controlling children’s central body fat deposition more effectively than in the CG. The mean adjusted difference of −1.5 cm observed in this study, in the changes of WC between the IG and the CG, is similar to the relevant difference of −1.7 cm, observed in the HopSCOTCH study. However, considering that the HopSCOTCH study was also conducted with a greater sample size (i.e., 107 children) and had a longer duration (i.e., 12 months), this probably highlights the promising potential of the tools that were developed and tested in this study, with regards to the effective management of childhood obesity.

The changes observed in the IG on BMI and WC, could be partly a reflection of the relevant favourable dietary changes recorded for the IG, compared to the CG. In this regard, the higher increase in dietary fibre intake in the IG than the CG and the significant decrease of dietary sucrose intake only in the IG are probably indicative of the effectiveness of the intervention in increasing the consumption of high-fibre foods that promote satiety and at the same time in decreasing the consumption of foods with a high sugar and, thus, high energy content. The aforementioned changes were also evidenced by the higher consumption of cereals at follow-up in the IG than the CG, as well as the significant decrease in the consumption of chocolates and cakes only in the IG. The above, in conjunction with the decrease in the consumption of chips in the IG, could possibly provide a basis that supports a lower dietary energy intake and consequently the favourable anthropometric changes observed for children in the IG. In line with the findings of the present study, the HopSCOTCH study also reported a higher diet quality score (reflected by the higher consumption of fruit, vegetables and water and by the lower consumption of fatty/sugary foods and non-diet sweet drinks) among 3–10-year-old obese children that received dietary and lifestyle optimisation advice for their weight management through a web-based software [28]. The fact that the HopSCOTCH study reported no significant differences between groups in the change of children’s physical activity levels from baseline to follow-up, indicates that any favourable changes observed in this study on the examined anthropometric indices are mainly attributed to the improvement of dietary habits in the intervention compared to the control treatment arm. To some extent, the same also applies in our study, as physical activity levels did not differentiate between the IG and the CG (data not shown).

Obesity in children has been strongly linked to important micronutrient insufficiencies, which is usually the outcome of a chronic, low-grade inflammation induced by the elevated levels of visceral adipose tissue [2]. As such, the DST was designed to assist children that received the personalised advice to achieve, not only a better management of their body weight, but also a higher intake of several essential micronutrients. This was evidenced by the significant increases in the dietary intakes of iron, magnesium and zinc observed only in the IG, which can correct potential obesity-related insufficiencies [3] and can subsequently support children’s growth, motor and cognitive function [29,30,31]. In addition, since hypertension is another common comorbidity of obesity in children [32], the dietary recommendations provided to children (particularly to those diagnosed with elevated blood pressure) and their parents via the DST, were also aiming to reduce the use of table salt, as well as the consumption of foods that are rich sources of salt in the diet. The significant decrease in dietary sodium intake observed in the present study only in the IG provides evidence that this additional aim of the intervention was partially achieved.

Our study has both strengths and limitations. The main strength was its randomised controlled design resulting in a homogeneity of children’s characteristics at baseline in both treatment arms. Another strength was the use of the DST to guide clinicians on effectively managing children’s elevated body weight, by accurately assessing their nutritional status and needs and by providing appropriate dietary and lifestyle optimisation advice to children and their families, encouraging family self-management of behavioural changes. As evidenced by the current and the STAR study [23], intervention approaches that involve self-guided behavioural changes by families may be better suited to sustain the intensity required for effective behavioural change than those that primarily rely on healthcare professionals to deliver the main bulk of the intervention [27]. In this context, the meal plans delivered by the health professionals to the families in the present study were only a guide for healthier eating and not a prescriptive pathway that was compulsory for the children and their families to follow. The emphasis was given mainly to the recommendations and how families can adopt and embed as many of these suggestions as possible to their daily life. Regarding additional strengths, according to qualitative feedback collected from the clinicians that used the DST, the paediatricians reported that the tool was quite easy to use (it runs with Microsoft Excel and/or Access) and represented a well-structured and quick procedure that helped them provide tailored advice to children and families. As far as limitations are concerned, although the study initially recruited 80 children, only 65 were examined at follow-up, resulting in a drop-out rate of approximately 19%. Nevertheless, the fact that only 5 out of 15 study participants that dropped out were originally allocated to the IG is an indication that the intervention was better accepted, increasing retention rates in the IG children and their families, compared to the CG that received only generic advice.

## 5. Conclusions

The current study showed that a computerised DST, designed to support paediatric healthcare professionals in the delivery of personalised diet and lifestyle optimisation advice to overweight or obese children and their families, resulted in improvement of the children’s dietary intake and BMI. These changes are indicative of the dynamics of the tool in supporting clinicians to improve the effectiveness of care. Interventions of longer duration and larger sample sizes are needed to confirm the findings of our study and to demonstrate their long-term sustainability.

## Figures and Tables

**Figure 1 nutrients-11-00706-f001:**
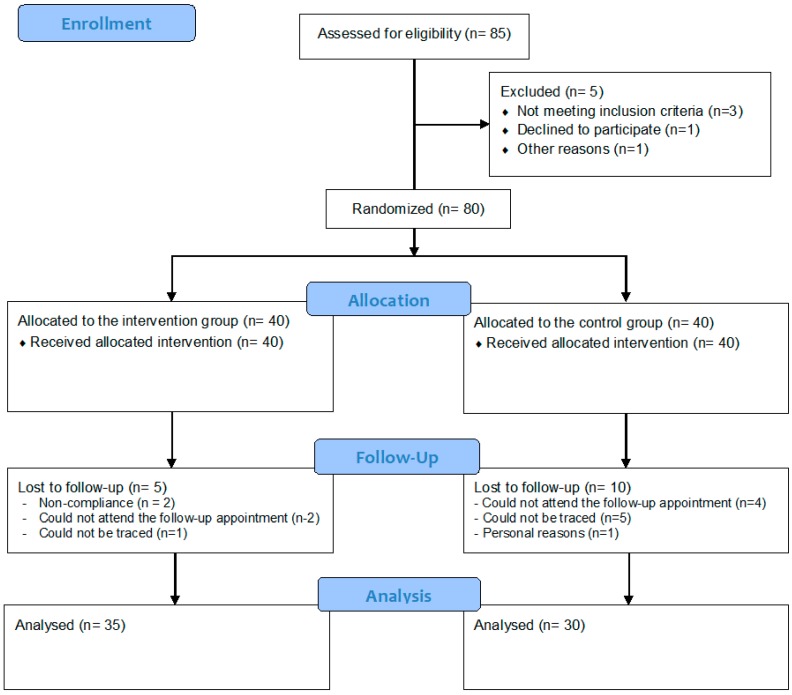
Flow diagram of study participants.

**Table 1 nutrients-11-00706-t001:** Population and behavioural change goals of diet and lifestyle optimisation advice provided through the decision support tool.

	Recommendation 1	Recommendation 2	Recommendation 3	Recommendation 4
***Target population *:***				
Children	☑		☑	
Children and Family		☑		☑
***Behavioural change goals***				
Keep a balanced diet, increase physical activity and Improve sleep habits	☑	☑	☑	☑
Increase consumption of foods rich in dietary fibre and complex carbohydrates			☑	☑
Reduce consumption of foods rich in simple sugars, total and saturated dietary fat, cholesterol and sodium			☑	☑

* Both the content and style of recommendations were adjusted to promote behavioural change to children only or to the entire family.

**Table 2 nutrients-11-00706-t002:** Descriptive characteristics of children and their parents at baseline.

	Total Sample (*n* = 65)	Intervention Group (*n* = 35)	Control Group (*n* = 30)	*p*-Value ^2^
***Data*^1^*on children***				
Age (years)	9.7 (0.2)	9.8 (0.3)	9.6 (0.2)	0.447
Dietary energy intake (kcal/day)	1535.6 (81.3)	1552.2 (65.6)	1548.3 (74.1)	0.969
Dietary protein intake (% of kcal)	18.5 (0.6)	18.3 (0.9)	19.2 (1.0)	0.511
Dietary carbohydrates intake (% of kcal)	47.4 (1.5)	47.1 (1.9)	46.3 (2.3)	0.790
Dietary fat intake (% of kcal)	35.4 (1.4)	36.2 (1.8)	35.6 (2.1)	0.840
Physical activity (min/day)	21.6 (2.3)	22.6 (3.0)	20.4 (3.5)	0.631
Birth weight (kg)	3.2 (0.1)	3.2 (0.1)	3.2 (0.1)	0.986
Recumbent length at birth (cm)	50.7 (0.4)	50.7 (0.5)	50.6 (0.6)	0.905
Body weight (kg)	51.9 (1.9)	54.3 (2.4)	48.6 (2.7)	0.127
BMI (kg/m^2^)	25.1 (0.5)	25.6 (0.7)	25.2 (0.7)	0.172
Overweight children (%)	39.3	42.4	35.7	0.593
Obese children (%)	60.7	57.6	64.3	0.593
Height (cm)	142.4 (1.4)	143.5 (1.9)	141.3 (2.0)	0.415
Waist circumference (cm)	79.9 (1.5)	81.0 (2.3)	78.3 (2.1)	0.388
***Data*^1^*on parents***				
Mother’s pre-pregnancy BMI (kg/m^2^)	24.9 (0.4)	23.8 (0.7)	26.2 (1.0)	0.055
Obese mothers before pregnancy (%)	15.5	9.1	24.0	0.163
Father’s age (years)	46.1 (0.3)	45.5 (0.8)	46.7 (1.0)	0.341
Mother’s age (years)	41.2 (0.3)	40.9 (0.9)	41.6 (1.0)	0.656
Mother’s education < 9 years (%)	24.6	24.2	25.0	0.535
Mother’s education > 12 years (%)	42.6	48.5	35.7	0.535
Father’s BMI (kg/m^2^)	28.6 (0.4)	29.1 (1.0)	28.1 (0.9)	0.452
Obese father (%)	27.6	37.5	15.4	0.084
Mother’s BMI (kg/m^2^)	27.3 (0.4)	26.0 (0.8)	28.9 (1.2)	**0.045**
Obese mother (%)	31.6	24.2	41.7	0.303

^1^ Data are presented as Mean (SEM) in the case of continuous variables and as percentages (%) in the case of categorical variables, ^2^
*p*-values derived from Student’s *t*-test or the non-parametric Mann-Whitney test in the case of continues variables and the Pearson chi-square test in the case of categorical variables. Figures in bold highlight statistically significant *p*-values.

**Table 3 nutrients-11-00706-t003:** Changes in dietary intake indices from baseline to follow-up.

	Baseline Mean (SEM)	Follow-Up Mean (SEM)	Mean Change (95% CI) (Time Effect)	*p*-Value ^†^
*Dietary energy intake (kcal/day)*				0.207
Intervention Group (*n* = 35)	1552.2 (65.6)	1467.6 (73.5)	−84.7 (−229.7 to 60.3)	
Control Group (*n* = 30)	1548.3 (74.1)	1605.9 (83.0)	57.6 (−106.3 to 221.5)	
*p*-value (Treatment effect)	0.969	0.225		
*Dietary protein intake (% of kcal)*				0.712
Intervention Group (*n* = 35)	18.3 (0.9)	17.5 (1.0)	−0.8 (−3.1 to 1.5)	
Control Group (*n* = 30)	19.2 (1.0)	19.1 (1.1)	−0.1 (−2.8 to 2.1)	
*p*-value	0.511	0.308		
*Dietary carbohydrates intake (% of kcal)*				0.777
Intervention Group (*n* = 35)	47.1 (1.9)	46.4 (1.6)	−0.7 (−4.5 to 3.1)	
Control Group (*n* = 30)	46.3 (2.3)	44.7 (1.9)	−1.6 (−6.1 to 2.9)	
*p*-value (Treatment effect)	0.790	0.514		
*Dietary fat intake (% of kcal)*				0.796
Intervention Group (*n* = 35)	37.5 (1.7)	36.2 (1.8)	−1.3 (−4.8 to 2.2)	
Control Group (*n* = 30)	35.6 (2.1)	37.7 (2.0)	2.1 (−2.1 to 6.2)	
*p*-value (Treatment effect)	0.840	0.953		
*Saturated fat intake (% of kcal)*				0.123
Intervention Group (*n* = 35)	13.0 (0.8)	12.6 (0.7)	−0.4 (−2.0 to 1.2)	
Control Group (*n* = 30)	12.9 (0.9)	14.4 (0.8)	1.5 (−0.3 to 3.3)	
*p*-value (Treatment effect)	0.887	0.099		
*Dietary cholesterol intake (mg/day)*				0.733
Intervention Group (*n* = 35)	288.9 (25.2)	245.1 (27.5)	−43.6 (−112.7 to 25.5)	
Control Group (*n* = 30)	239.0 (22.3)	211.5 (24.4)	−27.5 (−88.6 to 33.6)	
*p*-value (Treatment effect)	0.152	0.373		
*Dietary fibre intake (g/day)*				**0.047**
Intervention Group (*n* = 35)	13.0 (1.2)	17.1 (1.5)	**4.1 (1.4 to 6.8)**	
Control Group (*n* = 30)	11.9 (1.3)	12.0 (1.7)	0.2 (−2.9 to 3.3)	
*p*-value (Treatment effect)	0.534	**0.033**		
*Sucrose intake (g/day)*				0.680
Intervention Group (*n* = 35)	16.0 (2.4)	11.4 (1.6)	**−4.6 (−8.8 to −0.3)**	
Control Group (*n* = 30)	11.9 (2.7)	8.7 (1.8)	−3.2 (−8.0 to 1.7)	
*p*-value (Treatment effect)	0.279	0.267		
*Calcium intake (mg/day)*				0.067
Intervention Group (*n* = 35)	769.0 (143.6)	913.8 (60.1)	144.9 (−165.0 to 454.6)	
Control Group (*n* = 30)	1203.3 (162.4)	903.9 (67.9)	−299.3 (−649.5 to 50.9)	
*p*-value (Treatment effect)	0.053	0.915		
*Iron intake (mg/day)*				0.099
Intervention Group (*n* = 35)	10.7 (0.9)	13.3 (1.0)	**2.6 (0.2 to 5.0)**	
Control Group (*n* = 30)	11.9 (1.3)	11.4 (1.1)	−0.5 (−3.2 to 2.2)	
*p*-value (Treatment effect)	0.370	0.211		
*Potassium intake (mg/day)*				0.116
Intervention Group (*n* = 35)	1888.1 (141.6)	2052.6 (147.3)	169.5 (−110.4 to 449.4)	
Control Group (*n* = 30)	2119.2 (160.1)	1945.6 (166.5)	−173.6 (−490.0 to 142.7)	
*p*-value (Treatment effect)	0.292	0.622		
*Magnesium intake (mg/day)*				**0.011**
Intervention Group (*n* = 35)	192.1 (13.2)	228.7 (13.4)	**36.6 (9.4 to 63.8)**	
Control Group (*n* = 30)	228.0 (14.9)	209.8 (15.1)	−18.2 (−49.0 to 12.5)	
*p*-value (Treatment effect)	0.081	0.359		
*Zinc intake (mg/day)*				0.066
Intervention Group (*n* = 35)	7.6 (0.6)	9.3 (0.7)	**1.7 (0.1 to 3.3)**	
Control Group (*n* = 30)	9.6 (0.7)	9.0 (0.8)	−0.6 (−2.3 to 1.2)	
*p*-value (Treatment effect)	0.031	0.768		
*Sodium intake (mg/day)*				0.135
Intervention Group (*n* = 35)	1717.1 (210.4)	1426.2 (129.2)	**−290.9 (−745.4 to 163.7)**	
Control Group (*n* = 30)	1550.4 (237.8)	1788.3 (146.1)	238.0 (−275.9 to 751.8)	
*p*-value (Treatment effect)	0.608	0.073		
*Vitamin A intake (RE/day)*				0.137
Intervention Group (*n* = 35)	616.7 (122.9)	888.2 (243.5)	271.5 (−256.1 to 799.1)	
Control Group (*n* = 30)	670.4 (139.0)	330.5 (275.3)	−339.9 (−936.2 to 256.5)	
*p*-value (Treatment effect)	0.777	0.141		
*Vitamin C intake (μg/day)*				0.655
Intervention Group (*n* = 35)	81.5 (11.3)	60.9 (8.9)	−20.6 (−42.9 to 1.7)	
Control Group (*n* = 30)	70.2 (12.9)	41.9 (10.0)	**−28.4 (−53.6 to −3.1)**	
*p*-value (Treatment effect)	0.522	0.168		
*Vitamin D intake (IU/day)*				0.120
Intervention Group (*n* = 35)	93.2 (21.3)	107.9 (16.6)	14.6 (−29.2 to 58.4)	
Control Group (*n* = 30)	145.2 (24.1)	106.8 (18.7)	−38.4 (−87.9 to 11.1)	
*p*-value (Treatment effect)	0.117	0.965		

^†^*p*-values indicate the significance of the treatment × time interaction effects; adjustments were made for maternal BMI. Figures in bold highlight statistically significant *p*-values or statistically significant mean changes from baseline to follow-up.

**Table 4 nutrients-11-00706-t004:** Food intake from baseline to follow-up.

	Baseline Mean (SEM)	Follow-Up Mean (SEM)	Mean Change (95% CI) (Time Effect)	*p*-Value ^†^
*Fruits intake (portions/day)*				0.236
Intervention Group (*n* = 35)	1.14 (0.15)	1.26 (0.14)	0.12 (−0.18 to 0.41)	
Control Group (*n* = 30)	1.25 (0.18)	1.09 (0.17)	−0.16 (−0.52 to 0.19)	
*p*-value (Treatment effect)	0.643	0.455		
*Vegetables intake (portions/day)*				0.941
Intervention Group (*n* = 35)	0.94 (0.11)	0.93 (0.07)	−0.01 (−0.25 to 0.23)	
Control Group (*n* = 30)	0.87 (0.13)	0.88 (0.09)	0.03 (−0.27 to 0.29)	
*p*-value (Treatment effect)	0.701	0.665		
*Cereals intake (portions/day)*				0.446
Intervention Group (*n* = 35)	0.62 (0.10)	0.78 (0.11)	0.16 (−0.09 to 0.41)	
Control Group (*n* = 30)	0.42 (0.12)	0.43 (0.12)	−0.01 (−0.29 to 0.30)	
*p*-value (Treatment effect)	0.208	**0.041**		
*Fish intake (portions/day)*				0.502
Intervention Group (*n* = 35)	0.16 (0.02)	0.35 (0.11)	0.19 (−0.04 to 0.41)	
Control Group (*n* = 30)	0.15 (0.02)	0.22 (0.13)	0.07 (−0.20 to 0.33)	
*p*-value (Treatment effect)	0.565	0.440		
*Milk intake (portions/day)*				0.272
Intervention Group (*n* = 35)	1.05 (0.11)	1.17 (0.15)	0.11 (−0.21 to 0.43)	
Control Group (*n* = 30)	1.02 (0.17)	1.18 (0.13)	0.17 (−0.21 to 0.54)	
*p*-value (Treatment effect)	0.819	0.955		
*Yogurt intake (portions/day)*				**0.005**
Intervention Group (*n* = 35)	0.22 (0.07)	0.34 (0.04)	0.12 (−0.04 to 0.27)	
Control Group (*n* = 30)	0.50 (0.08)	0.26 (0.04)	**−0.23 (−0.42 to −0.50)**	
*p*-value (Treatment effect)	0.017	0.177		
*Chocolates intake (portions/day)*				**0.025**
Intervention Group (*n* = 35)	0.70 (0.09)	0.39 (0.06)	**−0.32 (−0.52 to −0.11)**	
Control Group (*n* = 30)	0.55 (0.10)	0.59 (0.08)	0.05 (−0.19 to 0.28)	
*p*-value (Treatment effect)	0.268	**0.044**		
*Fizzy drinks intake (portions/day)*				0.707
Intervention Group (*n* = 35)	0.08 (0.03)	0.08 (0.03)	0.004 (−0.05 to 0.06)	
Control Group (*n* = 30)	0.11 (0.03)	0.10 (0.03)	−0.01 (−0.08 to 0.06)	
*p*-value (Treatment effect)	0.366	0.676		
*Cakes intake (portions/day)*				0.317
Intervention Group (*n* = 35)	0.18 (0.05)	0.05 (0.01)	**−0.13 (−0.23 to −0.02)**	
Control Group (*n* = 30)	0.10 (0.07)	0.06 (0.01)	−0.04 (−0.17 to 0.08)	
*p*-value (Treatment effect)	0.268	**0.044**		
*Chips intake (portions/day)*				0.397
Intervention Group (*n* = 35)	0.14 (0.03)	0.06 (0.01)	**−0.08 (−0.13 to −0.03)**	
Control Group (*n* = 30)	0.09 (0.03)	0.04 (0.01)	−0.05 (−0.11 to 0.02)	
*p*-value (Treatment effect)	0.249	0.221		

^†^*p*-values indicate the significance of the treatment × time interaction effects; adjustments were made for maternal BMI. Figures in bold highlight statistically significant *p*-values or statistically significant mean changes from baseline to follow-up.

**Table 5 nutrients-11-00706-t005:** Anthropometric indices from baseline to follow-up.

	Baseline Mean (SEM)	Follow-Up Mean (SEM)	Mean Change (95% CI) (Time Effect)	*p*-Value ^†^
*Body weight (kg)*				0.360
Intervention Group (*n* = 35)	54.3 (2.4)	55.0 (2.4)	0.7 (−0.3 to 1.7)	
Control Group (*n* = 30)	48.6 (2.7)	50.0 (2.7)	**1.4 (0.3 to 2.6)**	
*p*-value (Treatment effect)	0.127	0.174		
*Height (cm)*				0.120
Intervention Group (*n* = 35)	143.5 (1.9)	145.5 (1.8)	**2.0 (1.5 to 2.5)**	
Control Group (*n* = 30)	141.3 (2.0)	142.7 (2.0)	**1.6 (1.0 to 2.1)**	
*p*-value (Treatment effect)	0.415	0.304		
*BMI (kg/m* ^2^ *)*				0.112
Intervention Group (*n* = 35)	25.6 (0.7)	25.2 (0.7)	**−0.4 (−0.9 to −0.1)**	
Control Group (*n* = 30)	24.1 (0.9)	24.3 (0.8)	0.2 (−0.4 to 0.8)	
*p*-value (Treatment effect)	0.172	0.389		
*BMI z-score*				0.318
Intervention Group (*n* = 35)	2.6 (0.2)	2.5 (0.1)	**−0.2 (−0.3 to 0.05)**	
Control Group (*n* = 30)	2.8 (0.2)	2.8 (0.2)	0.1 (−0.02 to 0.2)	
*p*-value (Treatment effect)				
*Waist circumference (cm)*				0.144
Intervention Group (*n* = 35)	81.0 (2.3)	81.6 (2.3)	0.6 (−0.9 to 2.1)	
Control Group (*n* = 30)	78.3 (2.1)	80.4 (2.1)	**2.1 (0.7 to 3.5)**	
*p*-value (Treatment effect)	0.388	0.705		

^†^*p*-values indicate the significance of the treatment × time interaction effects; adjustments were made for maternal BMI. Figures in bold highlight statistically significant *p*-values or statistically significant mean changes from baseline to follow-up.

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
