# Peer review of "Assessment of the Effectiveness of a Computerised Decision-Support Tool for Health Professionals for the Prevention and Treatment of Childhood Obesity. Results from a Randomised Controlled Trial"

_nutrients, 2019, doi:10.3390/nu11030706_

Round 1
Reviewer 1 Report
This manuscript describes the result of a randomized trial testing the effectiveness of a computerized decision-support tool (DST) in 65 families of 6-12 year old overweight or obese children. This is an important area of research in light of the vast need in controlling overweight/obese epidemic in many countries and prevention or management from young is certainly a very important approach. The authors have prepared the manuscript with good quality. The following lists some comments and suggestions for further refinement.
1. Throughout the manuscript, the authors analyzed and reported data of BMI. Since this is a growing population, why was not BMI for age percentile not considered for the analysis and report?
2. The description of the tool seems unnecessarily lengthy and with some repetition, would suggest edit to make it more succinct.
3. Please provide references for the FFQ and physical activity questionnaire (lines 311 and 322).
4. Figure 1, the numbers in the boxes showing exclusion reasons and lost to follow up don’t match up.
5. The authors reported and concluded in multiple places that IG group significantly reduced sucrose intake, however Table 3 shows no difference between arms.
6. As the authors pointed out that previous tools of similar design failed because of difficulty and challenges in using the tools. How was the usage by the clinicians assessed? How was the acceptance by the clinicians? How would the clinician evaluate the possibility of long term incorporation of this tool in clinical settings?
Author Response
This manuscript describes the result of a randomized trial testing the effectiveness of a computerized decision-support tool (DST) in 65 families of 6-12 year old overweight or obese children. This is an important area of research in light of the vast need in controlling overweight/obese epidemic in many countries and prevention or management from young is certainly a very important approach. The authors have prepared the manuscript with good quality. The following lists some comments and suggestions for further refinement.
1. Throughout the manuscript, the authors analyzed and reported data of BMI. Since this is a growing population, why was not BMI for age percentile not considered for the analysis and report?
Reply: The authors would like to thanks the reviewer for this comment. As suggested BMI for age z-scores have been calculated and presented in the revised version Table 5. Relevant additions have been incorporated in the methods (Page 7, Lines 431-432) and results (Page 13, Lines 553-554) sections of the manuscript.
2. The description of the tool seems unnecessarily lengthy and with some repetition, would suggest edit to make it more succinct.
Reply: The methods section that provides the description of the tool has been revised to become less lengthy and to avoid any unnecessary repetitions. Although the relevant text corresponding to the description of the tool has become more succinct, the fact that the description of the tool has not been previously published, this section still needs to remain as detailed as possible in order to also comply to the journal’s instructions for research articles, according to which:
“Materials and Methods: They should be described with sufficient detail to allow others to replicate and build on published results. New methods and protocols should be described in detail while well-established methods can be briefly described and appropriately cited.”
3. Please provide references for the FFQ and physical activity questionnaire (lines 311 and 322).
Reply: As suggested references for the FFQ and the physical activity questionnaire used in the present study have been embedded in the text (Line 409 and 421-422, respectively, at Page 7).
4. Figure 1, the numbers in the boxes showing exclusion reasons and lost to follow up don’t match up.
Reply: Figure 1 has been revised as recommended to facilitate consistency in the number of children excluded and lost follow-up with the relevant numbers listed next to the different reasons of exclusion or loss to follow-up respectively. Nevertheless, the authors would like to mention that the main reason for this inconsistencies were the formatting changes made by the journal on our submission, and specifically the fact that figure 1 was compressed between pages 8 and 9, thus leading to the coverage of information in the different parts/boxes of te flow diagram, which was the reason of the final inconsistencies in the numbers
5. The authors reported and concluded in multiple places that IG group significantly reduced sucrose intake, however Table 3 shows no difference between arms.
Reply: The authors would like to apologise for any possible confusion caused by this statement. Indeed the changes observed with regards to dietary sucrose intake were not significantly different between the IG and the CG. Nevertheless, the decrease in sucrose intake was indeed significant within the IG group, which is also of importance. For this reason, the text has been revised throughout the manuscript, so as to highlight that the decreases was observed only in the IG (i.e. in Lines 520, Page 10 & Line 611, Page 15).
6. As the authors pointed out that previous tools of similar design failed because of difficulty and challenges in using the tools. How was the usage by the clinicians assessed? How was the acceptance by the clinicians? How would the clinician evaluate the possibility of long term incorporation of this tool in clinical settings?
Reply: Based on qualitative feedback collected from the clinicians that used the DST, the paediatricians reported that the tool was quite easy to use and represented a well-structured and quick procedure that helped them provided tailored advice to children and families. This information has been incorporated in the strengths parts of the revised manuscript (Lines 654-659, Pages 15-16). Nevertheless, the possibility of the long term incorporation of this tool in clinical settings was not assessed in the current study, since this will be the focus of a study of longer duration that is planned to be implemented in due course.

Reviewer 2 Report
· This is a very interesting study of a very important topic. While the effect of the study is limited in the number of significant changes, it is still an impressive study. I think a few questions remain that can be addressed:
· This all seems fascinating and potentially develops a thorough care plan. I still am a little confused as to how detailed the plan is and what it might look like. Perhaps an example of the plan available online as a PDF would be helpful. I’m particularly interested in is the detail of the meal plans.
· I’m a little concerned that the plan is totally prescriptive and does not invite participation by the family in planning their intervention/goals. I think the authors should address that.
· How practical would it be for other centers to use this DST? What software is needed? Etc.
· How were the medical providers trained to use the DST? Is the training easy?
· What were the socio-demographics of the families? Income? Education?
· With the peri-natal factors, is it known whether the mothers breast fed?
· The caloric reduction is small at about 100 cals per day. How much does this differ from what the model would suggest?
· Since there were limited significant differences, it might be nice to calculate a power analysis to determine the number of children needed for a larger and more definitive study.
Author Response
This is a very interesting study of a very important topic. While the effect of the study is limited in the number of significant changes, it is still an impressive study. I think a few questions remain that can be addressed:
1. This all seems fascinating and potentially develops a thorough care plan. I still am a little confused as to how detailed the plan is and what it might look like. Perhaps an example of the plan available online as a PDF would be helpful. I’m particularly interested in is the detail of the meal plans.
Reply: The methods section that provides the description of the tool has been revised become less lengthy and avoid repetitions. As suggested by the reviewer, some typical examples of the recommendations and of the meal plans provided to children (and their families) have been translated and uploaded as supplementary material with the revised article.
2. I’m a little concerned that the plan is totally prescriptive and does not invite participation by the family in planning their intervention/goals. I think the authors should address that.
Reply: All families in the IG had the opportunity to discuss and contribute to the setting of their goals with the health professionals before receiving the recommendations and meal plans. In addition, as mentioned in the discussion section of the revised manuscript (Lines 522-525, Page 15), after providing the meal plans and lifestyle optimization recommendation (that as relevant as possible to the goals set by the family) the health professionals also encouraged a self-management of their behavioural changes. As evidenced by the other studies that followed a similar approach as the current one, interventions that involve self-guided behavioural changes by families may be better suited to sustain the intensity required for effective behavioural change. In this context, the meal plans delivered by the health professionals to the families via the DST, was only a guide of healthier eating and not a prescriptive pathway that was compulsory for the children and their families to follow. The emphasis was given mainly to the recommendations and how families can adopt and ember as many of these suggestions as possible to their daily life. A relevant statement has been added in the discussion section of the revised manuscript (Lines 650-654, Page 15).
3. How practical would it be for other centers to use this DST? What software is needed? Etc.
Reply: The DST runs with Microsoft Excel or Access. This information has been added in the revised version of the manuscript (Lines 656-657, Page 15). As such, it is very easy for any centre to use the DST, provided however that the recommendations and meal plans will be adjusted to the needs and characteristics of the target population (e.g. if the DST is used to another country than Greece).
4. How were the medical providers trained to use the DST? Is the training easy?
Reply: A manual of operation with detailed instructions on the use of the DST was prepared and distributed to medical practitioners prior to the commencement of the study. In addition, in both hospitals the medical providers were appropriately trained and instructed in the use of the DST by the dietitian, who was also present in all phases of the study (i.e. screening, baseline and follow-up examination), thus facilitating consistency in the execution of the study and the collection of data. This information has been added in the methods section of the revised manuscript (Lines 366-368, Page 6 ).
5. What were the socio-demographics of the families? Income? Education?
Reply: As suggested by the reviewer maternal education was added in Table 1 (no information was collected on income), which displays the descriptive characteristics of the study population. Based on this information, in the majority of study participants (42.6%) their mothers had > 12 years of education, although no differences were observed between the two study groups. A relevant statement has also been added in the results section of the revised manuscript (Lines 492-494, Page 9).
6. With the peri-natal factors, is it known whether the mothers breast fed?
Reply: Considering that breastfeeding was not part of the CORE index (i.e. the predictive that was incorporated in the DST to evaluate the future risk of childhood obesity), information on breastfeeding and its duration was not collected.
7. The caloric reduction is small at about 100 cals per day. How much does this differ from what the model would suggest?
Reply: The authors would like to thank the reviewer for this very meaningful comment. Considering that ~40% of children were overweight and as such they received isocaloric meal plans and recommendations that were targeting weight maintenance, while 60% of children were obese, thus receiving a caloric restriction of up to 300 Kcals in order to achieve a mild weight loss, the actual caloric reduction observed in the study seems reasonable. In this regard, the average caloric restriction suggested by the model was ~180 Kcals, which although slightly higher to the actual one, it seems to be reasonable considering limited compliance issues in children that most likely did not allow the absolute alignment between the actual and the proposed caloric restriction.
8. Since there were limited significant differences, it might be nice to calculate a power analysis to determine the number of children needed for a larger and more definitive study.
Reply: Based on results derived from a statistical power analysis, but also from information stemming from other similar studies, a sample of 172 children would be considered adequate to observe a weight loss of 0.3 BMI z-score units (80% power; a =0.05). A study of longer duration with larger sample size, as the one indicated above, is planned to be implemented in due course, following the adjustments of the DST based on the outcomes of this pilot study.

Round 2
Reviewer 2 Report
The authors have done excellent job of addressing all the reviewer concerns. The anuscript is ready for publication in its current form.